



# Application of DOPPLER SODAR in short-term forecasting of PM10 concentration in the air in the City of Krakow (PL)

Ewa Krajny[1], Leszek Osrodka[1] and Marek Wojtylak[2]

[1] IMWM-NR 40-045 Katowice, Bratkow 10
[2] Retired scientist (M.W.)

*Correspondence to*: Ewa Krajny (E.K.) ewa.krajny@imgw.pl, Leszek Osrodka (L.O.) leszek.osrodka@imgw.pl

**Abstract.** The article describes an attempt to apply data obtained from SODAR (Sound Detection and Ranging) poms for short-term forecasting of PM10 concentration levels in Krakow. Krakow is one of the most polluted cities in Central Europe (CE) in terms of PM10 concentration. The city authorities, based on the access to legal measures, have undertaken a number

of organizational and legal initiatives aimed at significantly improving air quality (e.g. ban on burning solid fuels for space heating, forecasting air quality (AQ) for the planned implementation of free public transport, etc.). At the same time, the unfavorable topographic location of the city reduces the possibility of natural ventilation. This article describes all these conditions, focusing on presenting a method for short-term correction of air quality for the planned implementation of free public transport. Currently, the forecast is being developed by the Institute of Meteorology and Water Management –

National Research Institute (IMWM-NRI) using the CALPUFF model, powered by meteorological forecast data of the mesoscale ALADIN model. The use of this model generally makes it possible to correctly predict the average daily concentration values; however, the maximum values are understated. Based on several years of measurements of the physical properties of the atmosphere using SODAR, the authors of the paper suggest that SODAR data can be considered for operational use to generate short-term forecasts.

**Keywords:** PM air quality forecast; SODAR

## 1 Background

Air pollution is a significant problem for residents of urban agglomerations. The growing size of cities and their growing number of inhabitants result in both an increase in emissions and a reduction in the efficient removal of pollutants outside the city limits. This then causes a periodic accumulation of pollution, which leads to the formation of smog episodes. There is

evidence that air pollution has a significant impact on both life expectancy and quality of life. During episodes, an increased number of hospital admissions and deaths are observed, especially among children and the elderly (WHO, 2004; Bell et al., 2001; Dockery et al., 1993). The occurrence of high concentrations of PM in Krakow is always as described below.

Increasing emissions as a result of lower air temperatures during the cold season, reducing the demand for heat and thus reducing emissions as a result of "warm winters" are well-known examples of emission control mechanisms resulting from

meteorological and climatological factors. Another and equally important role is played by thermal and dynamic conditions



in the boundary layer of the atmosphere, where - apart from solar radiation and dynamic factors - the degree of urbanization and land cover are also important (Xue et al., 2021; Ji et al., 2020; Xu i in., 2018; Engelbart i in., 2009; Fisher i in., 2006; Pringer i in., 2004; Arya, 1999). These issues have also been studied in Poland, where the problem of the high concentrations of particulate pollutants is particularly acute (Toczko, 2015; Ziemianski and Osrodka, 2012). Among the large Polish cities,

Krakow is the one that is the most polluted in terms of PM concentration. Therefore, research in this area is particularly intensive (Bajorek and Wezyk, 2016; Bokwa, 2010; Matuszko, 2007; Walczewski, 1994). For example, extensive research on secondary sources on the legacy and contemporary scientific works on climate diversity in Krakow can be found in Bokwa (2019) and Matuszko (2007).

The inspiration for the research was the analysis of the current state of knowledge regarding the atmospheric structure of the

border layer over Krakow (Godlowska et al., 2008; Lewinska et al., 1982; Morawska-Horawska, 1978). In connection with the introduction in 1993 in Krakow of the first automatic air quality monitoring system in Poland and the delivery of system equipment, including REMTECH Doppler SODAR delivered to the Center Observation Station of the Institute of Meteorology and Water Management – National Research Institute in Krakow-Czyzyny, the research became more extensive. On the basis of data obtained from SODAR, wind roses were developed for Krakow at an altitude of 50 to 550 m

at intervals of 100 m, and separately for different types of atmospheric equilibrium (Rozwoda, 1995), for mixing layers and wind circulating over the city (Fisher et al., 2006, Pronger et al., 2004) and for COST 720 (Engelbart et al., 2009), while the measurement results obtained at that time became a useful knowledge base on the subject at used in modern research (Godlowska and Kaszowski, 2019; Bajorek and Wezyk, 2016;). In recent years, thanks to the use of drones and balloons for civilian applications, as well as miniaturization of measuring instruments, experimental studies of the vertical structure of the

atmosphere from the border layer in Krakow have been widely developed, with an emphasis on the impact of meteorological conditions on the vertical profile of air pollutants (Sekula et al., 2021). These studies provide important information on the vertical distribution of PM10 concentrations under different meteorological conditions.

As part of research activities on the causes and effects of excessive air pollution, intensive corrective actions were carried out in Krakow. In Poland, legal solutions has been applied in accordance with the provisions of Directive 2008/50/EC of the

European Parliament and of the Council of 21 May 2008 on ambient air quality and cleaner air for Europe (2008). Therefore, in areas (zones) with excessive air pollution, it is necessary to establish air quality programs aimed at direction of corrective actions (Regulation of the Minister of the Environment, 2012). These programs, in addition to an important legislative function (they constitute an act of local law), are of significant preventive and educational importance. Due to the fact that the concentrations of PM10, PM2.5 and benzo(a)pyrene (B(a)P) in Krakow exceeded the limits specified in the applicable

legal standards, air quality programs have been implemented for several years. The effect of their implementation is, i.a., the so-called "anti-smog resolution" (Resolution No. XVIII/243/16, 2016), which from 2019 prohibits the combustion of solid fuels for space heating and results in a gradual reduction of PM pollution concentrations. As a result of these legislative





actions, the replacement of boilers for space and water heating, a process carried out since 1995, has increased sharply,
leading to a significant reduction in PM emissions in the municipal sector (the number of solid fuel boilers decreased by
around 15 000 boiler installations in 2017-2019) (AQP Malopolska, 2018). This, in turn, significantly reduced PM10
concentrations, shown in Figure 1 as the number of days when the PM concentration threshold was exceeded. Specific
meteorological conditions in recent years (mild and warm winters) have also contributed to the situation but have not
diminished the importance of the actual reduction of municipal emissions.

### PM10 [μg/m³] for the 2015-2020 period

**Figure 1** Total number of days with PM10 concentrations exceeding 50, 100 and 150 μg/m³ in the years 2015-2020

Failure to meet air quality standards in Krakow is also caused by the unfavorable geographical location of the city. In the
north and south, the historical center of the city, located in the Vistula valley, is sheltered by round hills that cross river
valleys. The biggest problem of Krakow is that from the west it is protected by elements of a topographically diverse area
called the "Krakow Bridge", and height of the area is 370 m above sea level (German, 2001). The hills of the Krakow Bridge
are a barrier for the entire city, reducing the efficiency of ventilation and limiting the clearance of the Vistula valley in the
western part of Krakow, threatening the influx of wind, with western air circulation dominating in Poland (**Błąd! Nie można
odnaleźć źródła odwołania.**). The location of Krakow in the Vistula Valley affects not only the anemological conditions but
also contributes to the formation of temperature inversions, determining a constant type of equilibrium of the atmosphere in
harsh conditions and reducing the ability to vertically mix air. Urbanization processes, which result in an increasing density
of taller and taller buildings, also contribute to the reduction of wind speed and the ability to remove air pollution outside the
city area. However, by reducing the horizontal dispersion of pollutants, urbanization often contributes to the improvement of



vertical ventilation conditions. This is facilitated by increased anthropogenic heat production, which reduces the frequency of inversions in areas with large communities.

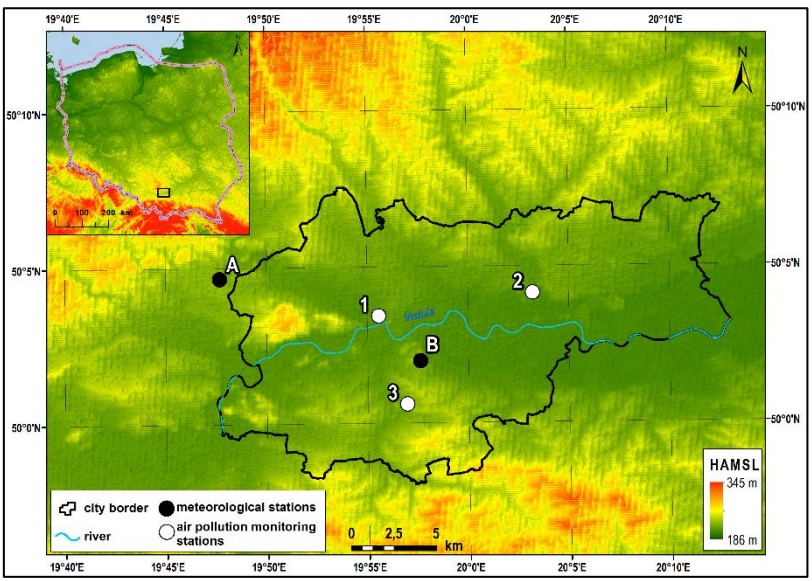

**Figure 2** Geographical location of the research area showing the locations of meteorological/air pollution monitoring stations. Meteorological stations of the Institute of Meteorology and Water Management – National Research Institute (marked in black): (A) Krakow-Balice, (B) SODAR; Measuring stations of the National Environmental Monitoring (NEM) (marked in white): (1) Krasinski Avenue, (2) Bulwarowa Street (3) Bujaka Street

## 2 Purpose of the study

The presence of episodes of high concentrations of PM10 in Krakow prompted the city authorities to develop a system that will provide residents with access to free public transport when the limit of 8 hours of concentration is to be exceeded the next day. The implementation of an air quality forecasting system for the control of public transport imposes the obligation to exercise the utmost care in modeling. A bad forecast is associated with either the risk of the city budget incurring unjustified costs (by multiplying the air quality forecast) or social (health) costs, which are difficult to estimate for an underestimated air quality forecast. In the specific conditions prevailing in Krakow, mainly local factors of stagnation, such situations are not uncommon. An additional element that makes it difficult to correctly forecast air quality using a deterministic model is the inability to provide an accurate emission inventory.

Given this state of affairs, methods are needed to frame air quality forecasts, in particular time trend forecasts and forecasts of the absolute value of PM10 concentration. The long-term activity of Doppler SODAR in Krakow and the analysis of the impact of atmospheric stability conditions on air quality gave the opportunity to use the results of these measurements to improve air quality forecasting (Bajorek and Wezyk, 2016).





# 3 Materials and methods

The work uses two data sources:

- Data from the selected Inspectorate for Environmental Protection/National Environmental Monitoring (IEP/NEM) (automatic air quality monitoring stations based in Krakow, data from 2015 to March 2022).
- MEASUREMENTS OF SODAR TAGS from the period 2017–March 2022).

The basic measurement characteristics and their locations are shown in Table 1 the station's code is as shown in Figure 2.

**Table 1**. Characteristics of measuring stations in Krakow.

| Monitoring network/ Owner/ Map symbol | Name/Location of the measuring station | Measured elements | Averaging time /Measurement type | Coordinates GPS | | Height m A.S.L. | Type of area/ Station type |
|---|---|---|---|---|---|---|---|
| | | | | Latitude φ N | Longitude λ E | | |
| No 1 | Krasinski Avenue | PM10 | 1 hour Automatic | 50°03′27.6" | 19°55′34.3" | 207 | urban/ communication |
| No 2 | Bulwarowa Street | PM10 | 1 hour Automatic | 50°04′09.5" | 20°03′12.6" | 195 | urban/ industrial |
| No 3 | Bujaka Street | PM10 | 1 hour Automatic | 50°00′38.1" | 19°56′57.1" | 223 | urban/ background |
| IMWM-NRI (A) | AMS Krakow-Balice | Meteorological parameters | 1 hour Automatic | 50°04′40″ | 19°47′42″ | 237 | extra-urban (suburban)/ background |
| Municipality of Krakow /MWM-NRI (B) | Swoszowicka Street SODAR (mobile station) | i.a.: vertical profile from 30 to 450 m above sea level, wind direction and speed characteristics, including atmospheric stability class DC (diffusion class) | 10 min Automatic | 50°02′0.63" | 19°57′37.85" | 247 | urban/ background |

## 3.1 Measurements of SODAR TAGS

The monostatic Doppler SODAR PCS.2000-24 system has been operating in Krakow since January 2015 for use in the implementation of the MONIT-AIR project. The system was manufactured by the German company METEK Meteorologische Messtechnik GmbH (https://metek.de).

The SODAR PCS.2000-24 system was built to measure wind speed and wind direction profile, on the basis of which turbulence parameters are determined in the form of atmospheric stability class and indirectly air temperature inversion. It is a monostatic Doppler SODAR system or transmitting antenna that transmits an audio signal and switches to receive and record a feedback signal. However, measurements using Doppler SODAR are based on the frequency of the Doppler return



signal as a function of time in relation to the transmitted signal. The monostatic SODAR PCS.2000-24 system transmits three audio beams simultaneously, each with three antennas: one vertical and two inclined phased antennas (Table 2).

The maximum range of vertical SODAR detection depends on the frequency of the transmitted signal. SODAR measures the
physical parameters of the atmosphere by analyzing the spectrum of sound waves dispersed by fluctuations in the atmosphere of different degrees, which are the result of heat and dynamic turbulence in the atmosphere (air temperature gradients, wind speed and hydrometeors). The Doppler shift for flat phased antennas depends only on the wind speed and the distance between the sound transmit-ting the sound. The technical specifications of SODAR are shown in Table 2 and an overview of SODAR is shown in Figure 3.

**Table 2**. Selected technical specifications of SODAR PC.2000-24 (Based on PSC.2000-24 tag manual, METEK version 2013)

| Parameter | Characteristics |
|---|---|
| Operating frequency | 1500... 2600 Hz |
| Range of measured speeds of horizontal wind elements | ±50 m/s |
| Range of measured wind directions | 0... 360° |
| Range of measured speeds of vertical wind elements | ±10 m/s |
| Minimum working height | ≥15 m (adjustable) |
| Minimum vertical resolution | ≥5 m (adjustable) |
| Height range | ~500 m (adjustable) |

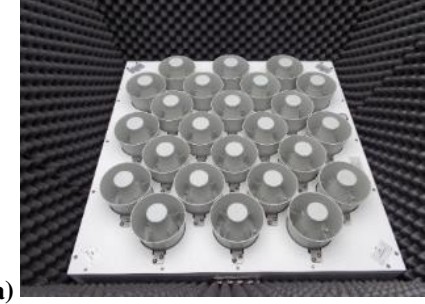
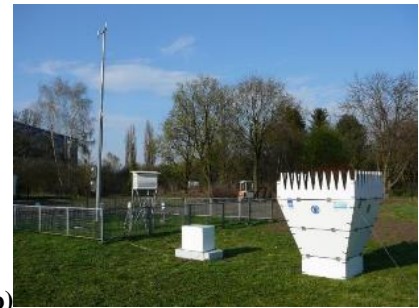

a)                                         b)

**Figure 3** METEK SODAR PC 2000-24: a) antenna panel with phased circuit, consisting of 24 speakers; b) Swoszowicka Street in Krakow
The methodology for compiling SODAR measurement results has been described i.a. by Emeis (2009) and Netzel et al.,
2000) and its usefulness is demonstrated, e.g. in Simakhin et al. (2017), Sgouros et al. (2011) and Yushkov et al. (2007). SODAR measurements are used for diagnostics or for setting physical property parameters in the atmospheric boundary layer (ABL). Other uses of SODAR were presented by Yang et al. (2020), who suggested using advanced SODAR data to forecast wind energy resources.

During the development of this study, SODAR was located in the vicinity of the Krzemionki Reservoirs at Swoszowicka
Street in Krakow. It should be mentioned that these SODAR locations represent a compromise between the presentability of





the measurement site for the largest possible area and the technical and organizational conditions required for the colocation of SODAR systems (Figure 2, Table 2).

The use of SODAR measurements makes it possible to determine several meteorological parameters and standards that can be directly used to analyze the air ventilation conditions in the city. These parameters include:

- wind speed and wind direction of the horizontal wind component (Ty and v);
- the wind speed and wind direction of the vertical wiatru component (w);
- atmospheric stability classes, which describe the state of stability of the atmosphere according to the modified Pasquill-Turner scheme and encoded in letters A to F (or numbers 1 to 6), where classes A, B, C (1 to 3) mean extremely, moderately and slightly unstable stability of the atmosphere; respectively, class D (4), means neutral
stability and classes E and F (5 and 6) mean stable and highly stable stability of the atmosphere.

All these elements were defined at levels up to 500 m with an approximate vertical resolution of 10 m. SODAR also provides raw data that determines the parameters of the transmitted and reflected signal and noise spectrum. These data were taken as input for the development of a PM10 forecasting algorithm based on SODAR data.

**3.2 SODAR-based PM10 prediction method**

**3.2.1 Justification for use**

The analysis of the occurrence of episodes of high concentrations of PM10 in Krakow and the classes of atmospheric stability identified by SODAR led to the idea of using SODAR data for short-term forecasting of PM10 concentrations in the city. Figure 4 illustrate example of an episode of high concentrations of PM10 dust on the background of DC determined from sodar.





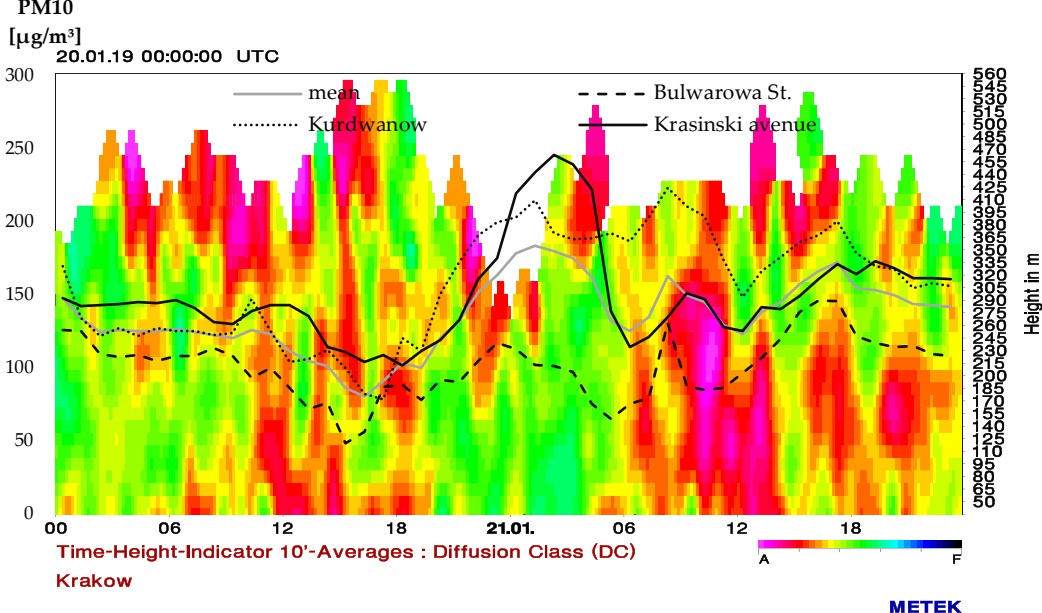


**Figure 4** The course of PM10 concentrations at selected Environmental Monitoring Stations in Krakow and the average value of measurements as at 20.01.2019, compared to atmospheric stability classes calculated on the basis of SODAR data

The lack of a visible relationship between the occurrence of high DC classes obtained from the visualization of SODAR data with the simultaneous high concentration of PM10 dust in stagnant weather conditions (high pressure system, low wind

speed, temperature inversion conditions) suggests that the information obtained from SODAR data only about DC is insufficient for a precise diagnosis and prediction of the development of an episode of high concentrations. PM10. In view of this state of affairs, it was decided to use properly processed data of the sodar spectrum, assuming that this would allow for a more complete analysis of ventilation conditions.

### 3.2.2 Preparation of SODAR data filtration

To build the PM10 concentration forecast, the basic results of SODAR measurements as a spectrum were used, i.e. a set of amplitudes of signals returning to the SODAR receiver from the reflection of sound transmission of a single frequency. Finding the required information in this way seemed to be the most promising option with regard to the purpose of the study (which was to analyze the ventilation conditions of air in the atmosphere). When a signal is reflected back through different layers of the atmosphere, it becomes scattered. The spectrum was recorded on 32 frequency channels around the trans-

mission frequencies. In a homogeneous atmosphere, the spectra should take the form of noise. To improve the transparency of the basic data of the SODAR spectrum, an initial transformation was required. After analyzing the spectrum for all altitudes and on different dates, it was found that as the altitude increased, the variability of the amplitudes of all frequencies in the spectrum decreased and a tendency to a nonzero, almost constant function appeared. In addition, there were secondary





maxima (external sound emitters) that could distort the analysis. It was also found that the shape of the spectrum was simi-

lar to the Gaussian curve. It was decided that the spectra should be simplified by subtracting their common part and similarly for spectra at all altitudes. This process is shown in Figure 5a-c. The first graph (Figure 5a) shows the original spectra measured at different heights, the second graph (Figure 5b) shows the spectra after subtracting their common part and the third graph (Figure 5c) shows the common part of all spectra called the "spectral background". Only the filtered spectra were transferred for further study. In addition, in order to slightly reduce the amount of data and adjust the frequency of measure-

ments of air volume, the spectrum of interest was averaged to one hour, although SODAR measures the data at 10-minute intervals.







**Figure 5** Example of FILTERING THE SODAR spectrum on 32 channels: (a) original spectrum; (b) spectrum with subtracted background; (c) context

### 3.2.3 Spectral properties of SODAR data

To Since it is difficult to compare full spectra with each other and with spectra from other periods, each spectrum is characterized by a single number (parameter). Thus arose the function of real values, the argument of which is height. This function will be called the "atmospheric state profile" (ASP). The most commonly observed feature of ASP is a rapid,



nonlinear decrease in value as the height increases. Attempts were made to determine as many parameters of the spectrum as possible. The numerical characteristics of the spectrum were chosen following a statistical approach, without analyzing their physical interpretations.

The characteristics in which ASP was determined were as follows:

A. Mean value of the SODAR reflection spectrum;

B. Maximum value of the spectra with a beam;

C. SNR (Signal-to-Noise Ratio);

D. Modal value (channel number with maximum high spectrum);

E. Standard deviation;

F. Median;

G. Skewness;

H. Kurtosis;

I. Similarity to the Gaussian curve.

Analyzing ASP shapes, their similarity to the Φ function turned out to be familiar (Eq. 1):

$$\Phi(h) = \frac{a}{h^2 + b} + c, \tag{1}$$

The ASP C profile was similar to another feature (Eq. 2):

$$\Psi(h) = a \cdot h^c \cdot e^{-b \cdot h} \tag{2}$$

Parameters a, b and c of equations (1) and (2) were determined for individual ASP (h – height above ground level using the quadratic mean approximation method). The medium-square approximation method determines the quality of the model's matching with the measurement data in the form of a coefficient of determination. This coefficient for transparency is marked RF – regularity factor and ranges from 0 (large irregularity) to 1 (perfect regularity).

The sample analysis suggested that RF = 0.373 is a bad regularity (Figure 6a) and RF = 0.956, which was a good regularity (Figure 6b). This half adjustment factor was treated as a factor of the regularity of ASP.

A general feature that was noticed was that the transformed spectra were close to zero from a certain height. Therefore, two versions of the calculation were carried out: up to 19 heights (215 m) and up to 28 heights (305 m).

Figure 7 shows that the ASP at 6 p.m. was irregular, meaning that the vertical structure of the atmosphere was deformed and irregular. ASP at 23:00 GMT was almost perfect. This led to the investigation of the relationship between the value of ASP and PM10 concentration. For example, the period from January 24 to 27, 2018 is shown in the chart in Figure 7 as the period during which excessive levels of PM10 concentrations occurred. The regularity factor for the characteristic was compared with the average concentration of PM10 from three automatic NEM stations near the SODAR location (on Krasinski Avenue, Bulwarowa and Bujaka Streets). For clarity, only one MF was shown, but the other regularity factors were similar.





After analyzing many ASPs, the hypothesis suggesting that ASP were disturbed a few hours ago, after which the concentration of PM10 increased, was introduced.

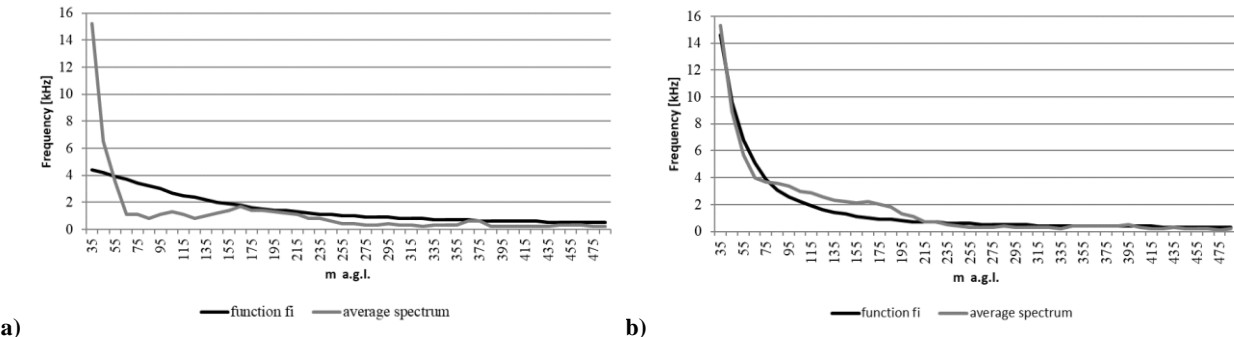

a)            b)

**Figure 6** Example of spectrum and SODAR *function Φ* for two cases: (a) irregularity to 18:00 UTC; (b) a good regularity to 23:00 UTC on 26 January 2018

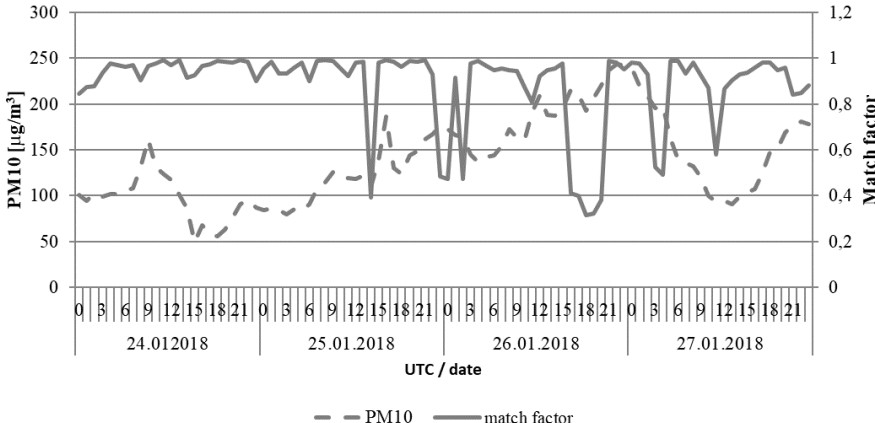


**Figure 7** Example of the regularity factor (RF) for characterization *a* (mean spectrum) between the SODAR data height value of 35-205 m and the PM10 concentration

For a more complete description of the state of the atmosphere, it was decided that the meteorological RF wind characteristics should be added to the regularity coefficients A to I:

J.   Horizontal wind speed (averaged in ASP);

     K.   Vertical wind speed (difference: min-max);

     L.   Wind direction (dispersion of direction around the average vector of this wind direction);

     M.  Temperature measured with SODAR at the level of 2 m above sea level.



## 4 PM10 forecast models

In the article we propose four methods of forecasting PM10 concentration in Krakow. Three of them use SODAR data, and the fourth is a reference method for what a forecast can be when only pollution data are available. The forecast is for the 12-hour time horizon, and the forecasting methods are based on SODAR and pollution data for the October-March winter seasons in the period 2017-2019. Reference data from the period from 1 October 2021 to 28 February 2022 were used to compare the methods. Each method was used to calculate the forecast every hour and its value was compared with the actual

(mathematical statistics: IEA, MAPE, MSE, UII Theila factor) measurement of PM10 concentrations. Thanks to this, a quantitative assessment of each of the methods was obtained and a correlation coefficient was derived for the relationship between PM10 measurement and short-term PM10 forecast.

Four methods for predicting PM10 concentrations have been developed, together with descriptions and application systems of the methods explained below (Figure 8).

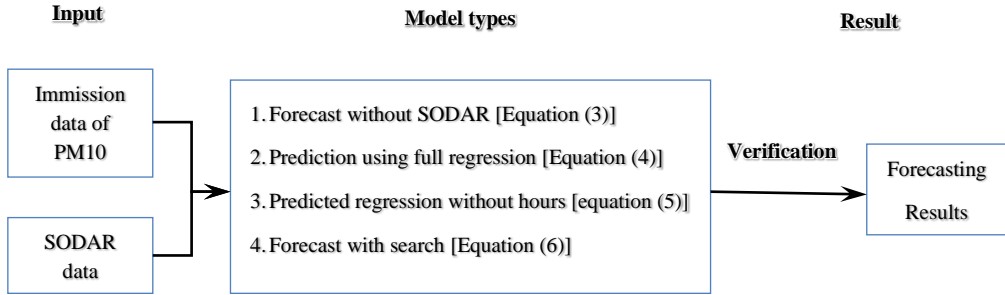


**Figure 8** SODAR-based forecast generation method

### 4.1 Reference method without SODAR data

Linear regression n was used to calculate the coefficients in equation (3):

$$PM10(h + 12) = b_0 + b_1 \cos\left(\frac{\pi \cdot h}{12}\right) + b_1 \cos\left(\frac{\pi \cdot h}{12}\right) + b_3 PM10(h), \tag{3}$$

where h is the hour (hour) of the day.

### 4.2 Full regression method

Linear regression was used to calculate the coefficients in equation (4):

$$ln\big(PM10(h + 12)\big) = b_0 + b_1 \cos\left(\frac{\pi \cdot h}{12}\right) + b_2 \sin\left(\frac{\pi \cdot h}{12}\right) + b_3 \ln\big(PM10(h)\big) + \sum_{ch=A}^{M} b_{ch} * MF^{ch}(h), \tag{4}$$

The corresponding transformations led to a weather pattern:





$$PM10(h + 12) = PM10(h)^{b_3} * e^{b_0 + b_1 \cos\left(\frac{\pi \cdot h}{12}\right) + b_2 \sin\left(\frac{\pi \cdot h}{12}\right) + \sum_{ch=A}^{M} b_{ch} * MF^{ch}(h)}.$$

**4.3 Hourless regression method**

In the full method, the arguments (independent variables) removed factors related to daily periodicity, since they also characterized SODAR factors. As a result, the following equations were formed:

$$\ln\left(PM10(h + 12)\right) = b_0 + b_1 \ln\left(PM10(h)\right) + \sum_{ch=A}^{M} b_{ch} * MF^{ch}(h). \tag{5}$$

The corresponding transformations have led to a forecasting pattern:

$$PM10(h + 12) = PM10(h)^{b_3} * e^{b_0 + b_1 \cos\left(\frac{\pi \cdot h}{12}\right) + b_2 \sin\left(\frac{\pi \cdot h}{12}\right) + \sum_{ch=A}^{M} b_{ch} * MF^{ch}(h)}.$$

**4.4 Search method**

There is a method in the form of an algorithm that is more flexible and can accommodate changes and improvements. The main idea of the method is to find the conditions in historical data (data mining, big data, big data form) that are closest to

the state of the atmosphere at the time of forecasting.

Historical data and current state are expressed as a vector of numbers:

$$X(h) = [PM10(h), MF^A(h), \dots, MF^M(h)]^T \epsilon \mathbf{R}^{14}. \tag{6}$$

The historical data for each X contains the actual values of PM10 concentration after 12 hours, which means that there is a sequence of pairs {(X (0), PM10 (12)), (X (1), PM10 (13)), ..., (X (n), PM10 (n + 12))}. Let Y denote the vector of the

current state. Y is related to X (0), ..., X (n), and the resulting extended sequence is normalized by transforming:

$$\tilde{p} = \frac{p - min}{max - min}, \tag{7}$$

where max and min are the maximum and minimum numbers for the corresponding coordinate.

The values of each coordinate are reduced to [0,1].

The sequence $\{\tilde{X}(0),\dots,\tilde{X}(n)\}$ is searched for the most similar vectors $\tilde{Y}$. To avoid the effects of the "dimensionality

curse", which means poor differentiation of radix in Euclidean spaces larger than 10, a fractional distance is used:

$$\rho(U, V) = \sqrt[k]{\sum_{i=1}^{n}(U_i - V_i)^k} \ \ U, V \epsilon \mathbf{R}^n, \tag{8}$$

where $k \in (0,1)$.

The most similar vectors are selected $X(k_1)$, ..., $X(k_s)$, and the value PM10 $(k_1 + 12)$, ..., PM10 $(k_s + 12)$ is used to generate a forecast for a situation 12 h after the current state. This may be an arithmetic mean, but in practice the geometric mean is

better. You can specify $k$ and $s$ values at any time. For the purposes of this test, $k = 0,6$.





Let's mark:

$$r_{min} = \min_{h \epsilon H} \rho(\tilde{Y} - \tilde{X}(h)), \tag{9}$$

Let the element $\tilde{X}(h_{min})$ denote the element that realizes this minimum. PM10 (hmin + 12) can be taken as a 12-hour forecast in the current situation. However, such a choice is very ineffective. That is why we propose a statistical approach.

Guided by inequality:

$$\rho(Y^{\tilde{}} - X^{\tilde{}}(h)) < (1 + \tau)r\_min, \tag{10}$$

select for the sample the values of PM10 (h + 12) for the periods in which the inequality is satisfied. On the basis of the sample obtained in this way, various descriptive statistics can be calculated. This may be an arithmetic mean, in practice the geometric mean was better. In addition, good results were obtained for the median or quantiles Q (0.35) and Q (0.6). You $\tau$

can specify k values at any time. For the purposes of the study, k = 0.6, $\tau$ = 0.5 was assumed. The sample size was usually around 100.

The undoubted advantage of this method is the constant supplementation of historical data. After 12 hours, when the value of the concentration of PM10 is known, the current state (situation) is considered historical, and further forecasts are driven by this latest historical data. This method also uses the division of modeling results into results for the geometric mean and

those for the median.

### 4.5 Results

Forecasting methods were verified for the entire population of areas with an average concentration of PM10 in Krakow in the period from October 2021 to March 2022 and episodes of high concentration of PM10 (events during which the momentary concentration of PM10 was above 100 µg/m$^3$). For the initial assessment of the forecast, it was compared with a

situation in which the forecast would be replaced by an average value (option 0). To verify the correctness of the forecasting, the first step was to determine the forecast errors for the entire data population using the following metrics:

MAE – Average Absolute Error:

$$MAE = \frac{\sum_{t=1}^{T}|y_t - Y_t^*|}{T}, \tag{11}$$

MAPE – Average absolute percentage error:

$$MAPE = \frac{\sum_{t=1}^{T}\frac{|y_t - Y_t^*|}{|y_t|}}{T} * 100\%, \tag{12}$$

MSE – Average Square Error:





$$MSE = \frac{\sum_{t=1}^{T}(y_t - Y_t^*)^2}{T},$$ (13)

UII – Theila factor:

$$UII = \frac{\sum_{t=1}^{T}(y_t - Y_t^*)^2}{\sum_{t=1}^{T} y_t^2},$$ (14)

where T is the sample size (forecast length); $y_t$ represents the measurement value; $Y_t^*$ denotes the measurement forecast and

CORR is a correlation factor.

**Table 3**. Basic statistical characteristics of the differences between forecasts and PM10 measurements for the entire data
population in the period October 2021 – March 2022

| Statistical Parameters | Measurement vs. | | | | | |
|---|---|---|---|---|---|---|
| | Average Option 0 | No SODAR Method (3) | Full Regression Method (4) | No hours Regression Method (5) | Search on Geometric Mean Method (6) | Median search |
| IEA (7) | 20.8 | 15.87 | 16.09 | 16.15 | 15.76 | 16.16 |
| MAPE (8) | 97.1 | 65 | 67.2 | 68.2 | 67.9 | 69.4 |
| MSE (9) | 727 | 477 | 474 | 475 | 445 | 471 |
| UII (10) | 0.326 | 0.214 | 0.212 | 0.213 | 0.199 | 0.211 |
| CORR | 0 | 0.591 | 0.657 | 0.648 | 0.636 | 0.626 |

The results obtained for the whole population revealed that for all forecasting methods, the error results were better than if

the average value was used (Table 3). This means that the forecasts have been made as intended. Analysis of specific model

data revealed various statistics on forecast errors, while the IEA (7) achieved the lowest values for the prediction of the

search method using a geometric mean (15.76). MAPE (8) had the lowest value for the be reference method from the no

SODAR data (65%). MSE (9) and UII (10) proved that the search method using geometric mean was the best. CORR

analysis showed that it was best with the full regression method. The results of such analyses for the whole population did

not show a preference for a single forecasting method.

In the second stage, episodes of high concentration of PM10 were distinguished for detailed analysis. Episodes of high

concentration of PM10 are critical situations for the implementation of free public transport in Krakow (and where PM10 >

100 μg/m³).

In the period from October 2021 to March 2022, four episodes were recorded on the following dates: 12-16 December 2021;

25-30 December 2021; 23-26 January 2022; 13-16 February 2022. The trend in the observed levels of PM10 concentration

and the projections of these bases on four interesting methods are shown in Figures from 9 to 12.





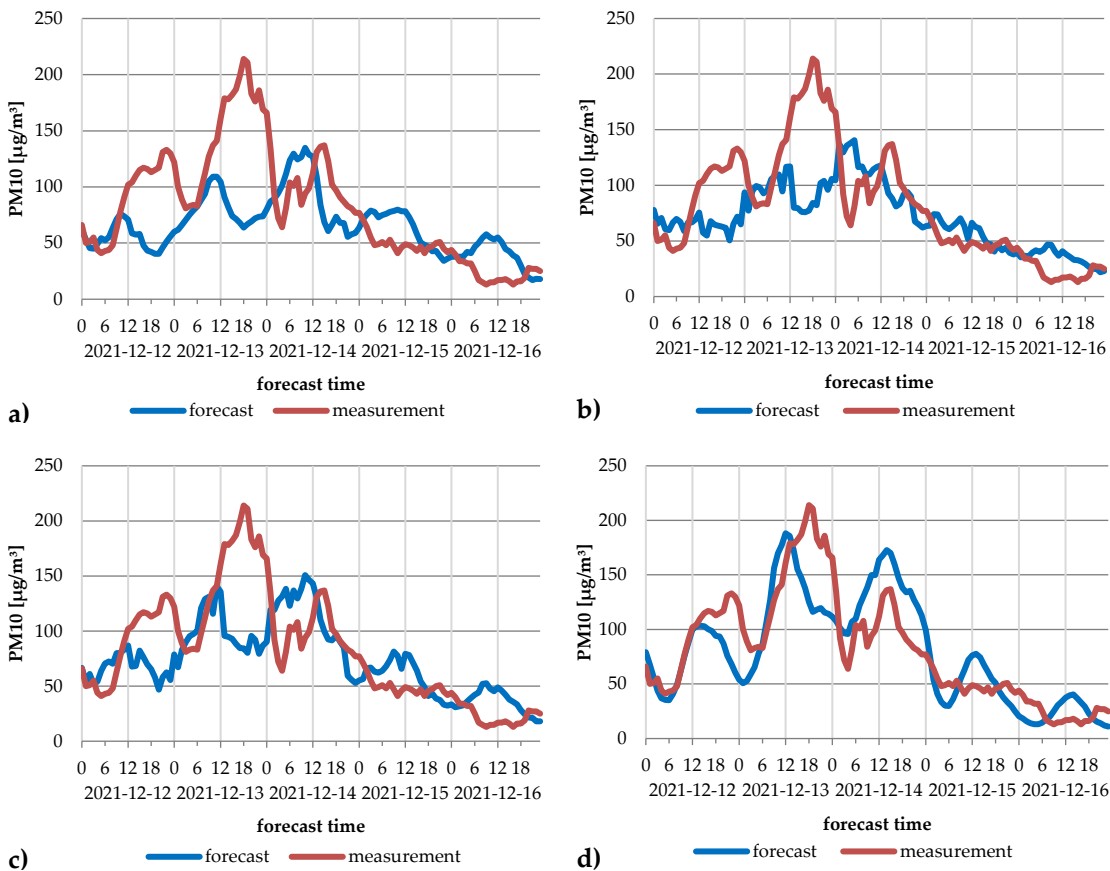

**Figure 9** Measurements and forecasts for the period from 12 to 16 December 2021: (a) forecast without SODAR data; (b)
predicted by regression without hours; (c) a forecast of full regression; (d) a forecast by means of a search method





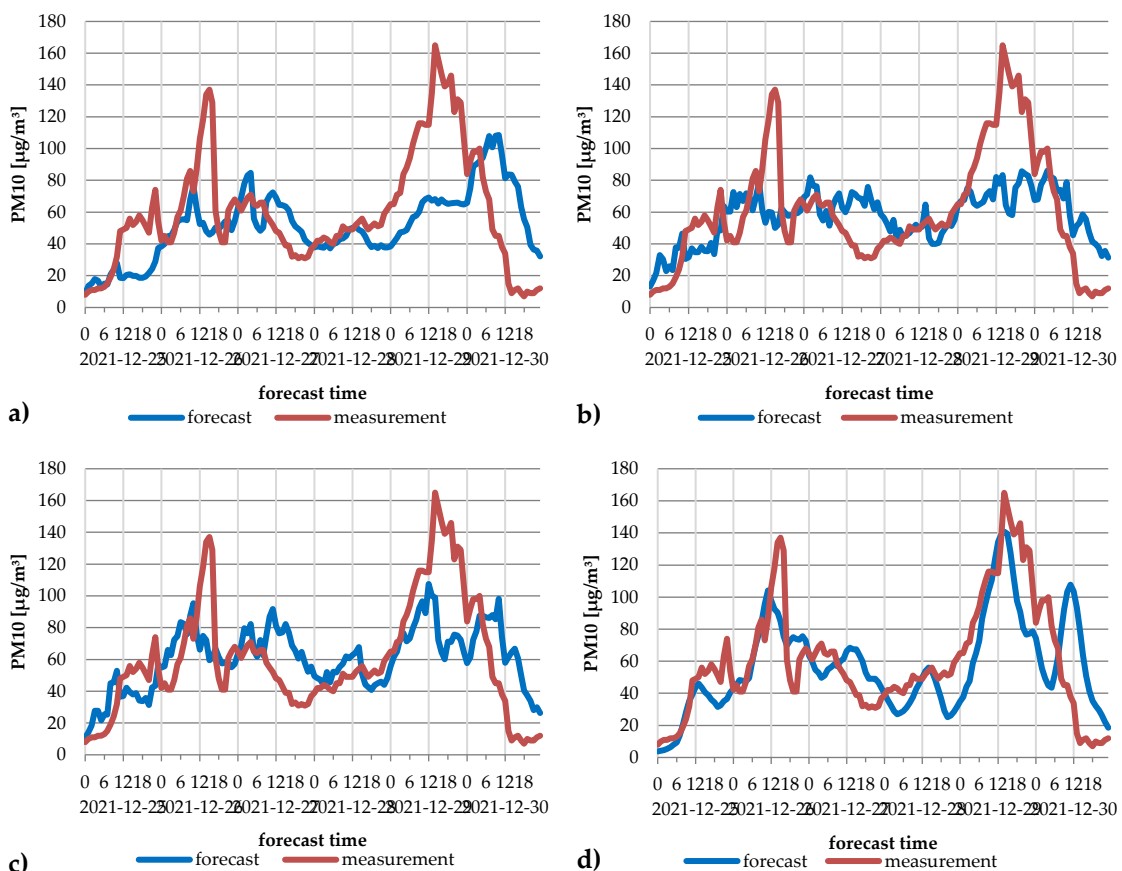

**Figure 10** Measurements and forecasts for the period from 25 to 30 December 2021: (a) forecast without SODAR data; (b) predicted by regression without hours; (c) a forecast of full regression; (d) a forecast will support the search method





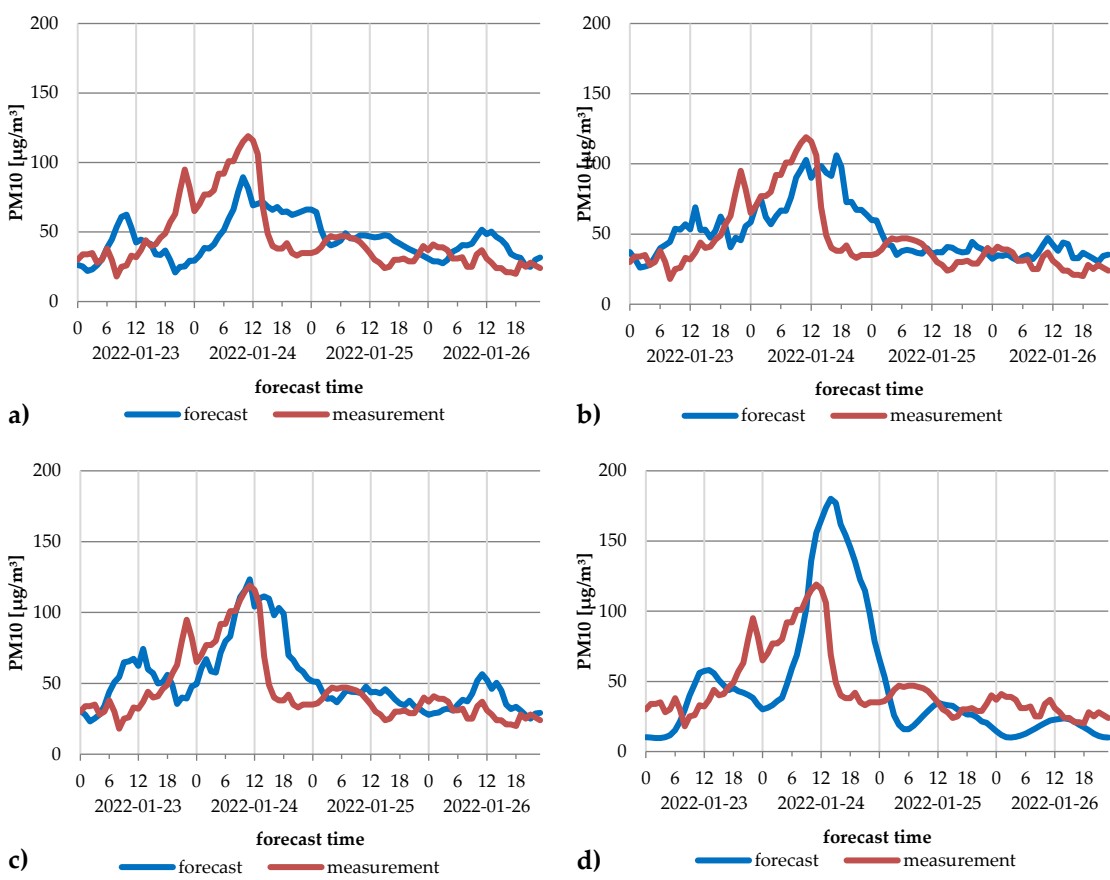

**Figure 11** Measurements and forecasts for the period from 23 to 26 January 2022: (a) forecast without SODAR data; (b) predicted by regression without hours; (c) a forecast of full regression; (d) a forecast by means of a search method





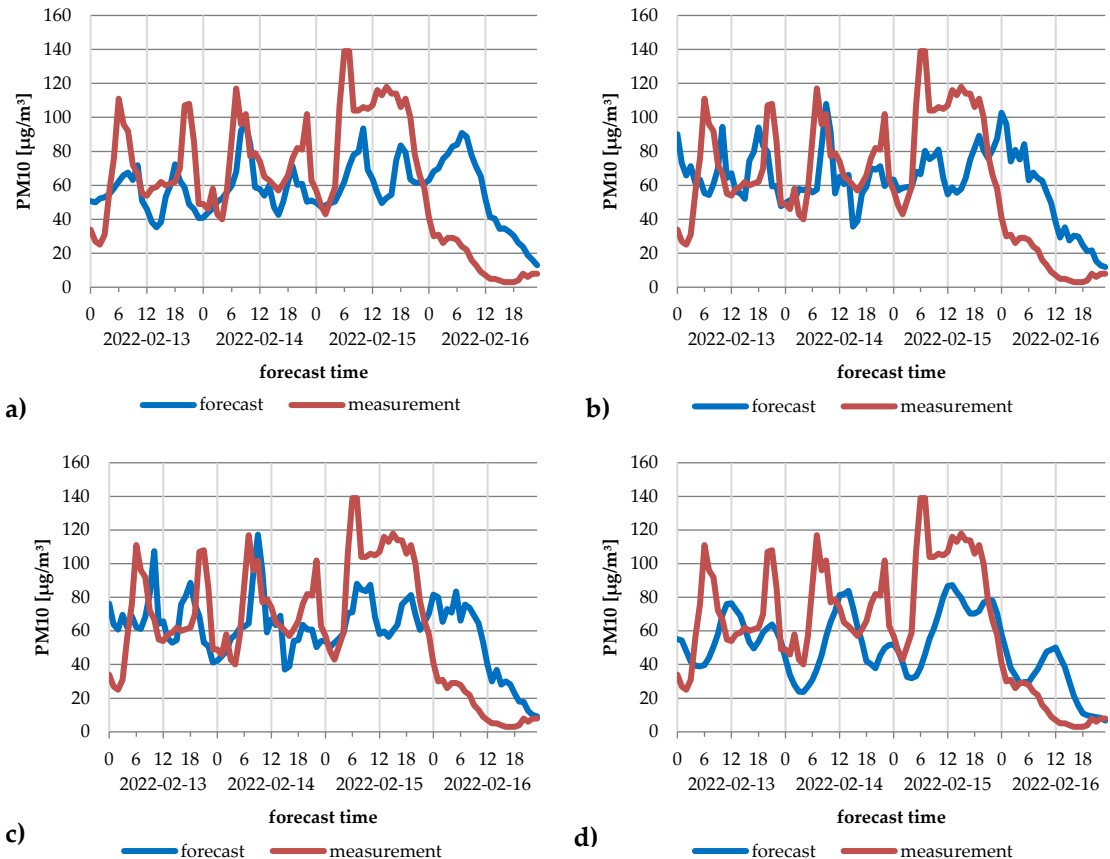

**Figure 12** Measurements and forecasts for the period from 13 to 16 February 2022: (a) forecast without SODAR data; (b) predicted by regression without hours; (c) a forecast of full regression; (d) a forecast by means of a search method

The above trends in PM10 predictions compared to measurements during PM10 episodes > 100 μg/m$^3$ (Figures from 9 to 12) revealed that, overall, each prediction method underestimated the measured maximum PM10 concentration. It should also be noted that the discrepancy between predictions and measurements changed with episodes. It can be concluded that the meteorological origin of each of the episodes was different, which made forecasting difficult. Despite these discrepancies, it was found that each of the methods of prognozation provided a trend result that mimicked the measurements, sometimes

with a delay of several hours. There were situations and types of methods that fit well. This was the case, for example, for the forecast (d) for the section 12–16 December 2021 (Figure 9d), forecast (d) for the section 25–30 December 2021 (Figure 10d) and the forecast (c) for the section from 23–26 January 2022 (Figure 11c).





**Table 4.** Basic statistical characteristics of the differences between PM10 projections and measurements for PM10 episodes > 100 μg/m³ in the period from October 2021 to March 2022

| Statistical Parameters | Measurement vs. | | | | | |
|---|---|---|---|---|---|---|
| | Average Option 0 | No SODAR Method (3) | Full Regression Method (4) | Regression No hours Method (5) | Search on Geometric Mean Method (6) | Median search |
| IEA (7) | 33.35 | 26.61 | 23.46 | 22.95 | 24.16 | 23.89 |
| MAPE (8) | 74.1 | 70.7 | 62.9 | 60.8 | 60.4 | 61.6 |
| MSE (9) | 2207 | 1343 | 1013 | 1010 | 1141 | 1113 |
| UII (10) | 0.397 | 0.242 | 0.182 | 0.182 | 0.205 | 0.2 |
| CORR | 0 | 0.475 | 0.62 | 0.627 | 0.599 | 0.593 |

The aggregated characterization of the forecast error statistics for episodes (Table 4) revealed that, as with the entire data population, the use of each forecasting method produced better results than the use of the average, which means that the predictions were useful. As for the statistical parameters studied, the method (5) – regression without hours (with the lowest MAE, UII and MSE and the highest CORR) was the best fit. However, this does not mean that this method is universal or suitable for every episode case, as shown above.

## 5 Conclusion


The use of PM10 forecasts for use in short-term activities undertaken to improve air quality is becoming more and more frequent in Poland (in three-day forecasts and in forecasts of the Chief Inspectorate for Environmental Protection, available online: http://powietrze.gios.gov.pl/pjp/airPollution). However, air quality forecasting is rarely used to guide administrative and economic decision-making (as in the case of implementing free public transport). This is due to the fact that inaccurate

forecasts cause high social costs (dissatisfaction of residents) or unjustified financial costs (lack of revenue from tickets). Therefore, applying air quality models to these forecasts must ensure that there is as little loss as possible due to poor decision-making.

The research discussed here shows that using SODAR data to support an air quality forecasting system is reasonable. In particular, the following proposals were made:

• The SODAR model can be a complementary for other forecasting methods, as it is very useful due to its simplicity and speed of calculations.

• The SODAR model does not require emission data, for which temporal and spatial variability is difficult to quickly verify.

• Table 4 shows that, especially at high concentrations, SODAR data provide significant information relative to the

model (3) without SODAR.



- The use of simple formulas for regression models in forecasting, while maintaining their multivariance (taking into account the four forecast options), facilitates the optimization of the predictive process.
- The model is ready for use, but work is underway to improve it through a different selection of sodar parameters.

Applying the model proposed in this article may improve short-term predictions of air quality, although the model still requires further testing, especially for episodes of high PM10 concentrations.

**Author Contributions:** Conceptualisation, E.K. and L.O.; methodology, E.K.; software, M.W.; validation, M.W. and E.K.; formal analysis, L.O.; investigation, L.O.; resources, E.K.; data curation, E.K.; writing – original draft preparation, L.O.; writing – review and editing, E.K.; visualisation, E.K.; supervision, M.W.; project administration, L.O.; funding acquisition, L.O. All authors have read and agreed to the published version of the manuscript.

**Funding:** This paper was also developed from the research results produced by the AIR BORDER project, funded by the Ministry of Science and Higher Education of the Republic of Poland, Contract No. 3911/INTERREG VA/2018/2. The research was co-financed from the science fund under INTERREG CE AIR TRITIA, and by the Ministry of Science and Higher Education of the Republic of Poland (Contract No. CE1101). The MONIT-AIR project was co-financed with funds from the European Economic Area's 2009-2014 Financial Mechanism.

**Institutional Review Board Statement:** Not applicable.

**Informed Consent Statement:** Not applicable.

**Data Availability Statement:** The meteorological data used in this paper (the wind roses) are available to the public: https://danepubliczne.imgw.pl/ (accessed on 5 July 2022). The air quality data used in this paper are available to the public: https://powietrze.gios.gov.pl/pjp/archives (accessed on 1 July 2022). The SODAR measurement data are available on request; the data are not available to the public as they are a collection of working data.

**Conflicts of Interest:** The authors declare no conflicts of interest.

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
