# Peer review of "Application of DOPPLER SODAR in short-term forecasting of PM10 concentration in the air in Krakow (Poland)"

_Atmospheric Measurement Techniques, 2023_

## Author Comment (AC2)

https://doi.org/10.5194/amt-2023-116-RC2

Dear Reviewer,

On behalf of the Authors, we would like to thank you for your valuable comments and comments. These were included in the revised manuscript

*The Author discuss the use of Doppler sodar data to inform a PM10 forecasting model for the city of Krakow. A number of regression methods involving sodar data are proposed and evaluated with independent PM10 measurements made at air quality stations. In most cases the sodar data seem to produce some improvement in the predicted PM10 levels compared to a regression method not using such data. In general the paper needs a major language revision. In particular the explanation of the methods in Section 3.2 needs to be made more understandable. When it comes to the evaluation of the impact of using sodar data as an input for regression, it may be useful to look at how the metrics for the sodar-based regression differ to those of the one not using sodar data on a case-by-case basis. My impression is that the impact of the sodar data varies quite a lot from situation to situation.*

*MAIN COMMENTS*

*Section 3.2.3, which describes the core of the methodology, was the hardest to understand for me.*

*- L187-202. "Thus arose the function of real values, the argument of which is height", etc. I do not get what exactly is being done. Are you parameterising each of the quantities in the bullet points A, B, C, etc., as functions of h based on eqs. 1 and 2?*

You get the point. Each spectrum coming from a given height is 32 numbers. Therefore, we try to characterize this 32-element sequence with a single value, e.g. an average. For each height (spectrum) we have one number, i.e. we get a function whose arguments are heights and ordinates are the values of a single characteristic. In addition, this function is called the ASP profile and is denoted by one of the capital letters A, B,...,I.

*- Eqs. 1 and 2. What do these functions represent? Do you mean that each quantity in the points A-I is parameterised as Phi (h) with suitably chosen a, b and c using a least squares fit? What do you mean by "C profile"? Is the "regularity factor" just the correlation coefficient between data and model?*

Exactly. Profiles A, B, D, E, F, G, H, I are approximated using the $\Phi$ function. Type C profiles have a different shape and are approximated by the $\Psi$ function. The functions $\Phi$ and $\Psi$ were chosen arbitrarily. The correlation coefficient squared $(r)^2$

*- Eq. 4. What is MF? By comparison with eq. 3 I guess this somehow represents sodar data, but which variable do you actually use in the regression?*

Evident error, it should be RF. Fixed

*- Eq. 4-5. Here you show a regression equation for PM10(h+12) and one for its logarithm. Which one do you ultimately use? Or do you somehow use both?*

This is the same equation in equivalent forms. A character from line 261 is used. The figure of line 259 is the result of linear regression.

*- L254. What do you mean by "the corresponding tranformations led to a weather pattern"?*

Of course, not the weather, but forecasting

*- L260. Same question, just with "forecasting" instead of "weather"*

By forecasting pattern, we mean formulas directly used to calculate the forecast;

*- As I mentioned in the preamble, it may be worthwhile to discuss Figures from 9 to 12 in some more detail. For example, if I look at Fig. 9 and 10 I see a clear impact of sodar data in following the observed PM10 concentrations. If I look at Figure 12, the impact looks less clear, and I am not sure the sodar-based forecast are significantly different to those not using sodar. Can you elaborate a bit more on what the reasons for this could be?*

It is obvious that the current state of the atmosphere may have no effect on the state in 12 hours, but we have observed such a relationship quite often. The most important advantage of our forecasting is its ease and immediacy. The data comes from a single source (sodar) and the forecast comes down to substituting it into one formula. We get the forecast almost immediately after reading the sodar data.

*- Are the regression coefficients determined from the same data as those plotted in Figs. 9-12 or do they come from independent measurements?*

Regression coefficients were determined on the basis of data from 2017-2019. Examples of use are from 2021-2022

*MINOR / TECHNICAL COMMENTS*

*- L7, "poms". Possible typo? Could understand what word was meant to be there.*

Fixed

*- L77-78. There is some sort of error message in Polish, maybe a missing reference to some figure or table.*

Fixed

*- L141, "wiatru" -> "wind"? (I looked it up on a translator)*

Fixed

*- L165, "the basic results... were used". So, is this a sodar Doppler spectrum?*

Yes, it is a Doppler spectrum

*- L175. How is the "common part" defined? Is it the mean of the spectra at all altitudes?*

Yes

*- L315, "This means that the forecasts have been made as intended". I guess here you mean that your forecast has some added value as it does better than just using the mean value as a guess.*

That's your perfect reasoning.

*- L325, "the projections of these based.." -> "the predicted values based on the four considered methods..."*

Language-corrected wording

On behalf of all authors

Leszek Ośródka

Katowice, 27 December 2023

---

## Author Comment (AC3)

https://doi.org/10.5194/amt-2023-116-RC1

**Dear Reviewer,**

On behalf of the Authors, we would like to thank you for your valuable comments and comments. These were included in the revised manuscript

The article presents three methods of short-term forecasts of PM10 concentrations for the 12h time horizon using the transformed Sodar spectra and concentration measurement results from three monitoring stations in Krakow, Poland. The algorithms were developed by the authors for 1h averaged data from the winter periods of 2017-2019, and then verified for the entire October 2021-March 2022 period and episodes of high PM10 concentrations at that time. For comparison, the authors also used the reference method without Sodar data. Satisfactory results were obtained, the authors suggest that the research should be continued with a different selection of Sodar parameters. From the formal point of view, the manuscript provides a new approach using original sodar data to forecast 1-hour PM10 concentrations for the city of Krakow. The authors correctly interpret the research results and present justified conclusions.

However, the manuscript has a number of editorial ambiguities that require clarification before the manuscript would be considered for publication. Below, there are main questions in order to make the manuscript clearer and more concise.

**General remarks**

**Abstract**

The abstract is not clear and concise. After a brief introduction of the topic, it should conain the key points of the article . There is a lack of presentation of research methods and major achievements. Some information should be included in the Introduction (I. 9 to 16).

We agree with the suggestions of the author of the review - the abstract will be edited in a way that includes the key points of the article and the general information will be moved to the introduction.

**Background**

This part should be more connected with the aim of the study. The authors generally presented the research conducted on the vertical structure of the atmosphere over Krakow, but did not discuss the models currently used for short-term forecasts of PM10 concentrations, which was the purpose of the work.

Suggestion accepted. In the final version of the article, a paragraph will be added about the currently used methods of short-term PM10 concentrations in Krakow.

**Purpose of the study**

The purpose of the research should be clearly formulated - currently it is a bit blurry.

My suggestion is. for example: We used " properly processed data of the sodar spectrum, assuming that this would allow for a more complete analysis of ventilation conditions ….", and further for what purpose

or "To build the PM10 concentration forecast, the basic results of SODAR measurements as a spectrum were used, i.e. a set of amplitudes of signals returning to the SODAR receiver from the reflection of sound transmission of a single frequency". Three method were developed....., etc..

*I believe, although English is not my native language, that the text needs to be revised for linguistic correctness.*

Suggestion accepted - the purpose of the research will be more precisely defined

**Specific remarks**

39-40 "The inspiration for the research was the analysis of the current state of knowledge regarding the atmospheric structure of the border layer over Krakow" - these studies may not be the latest (1978, 2008) and please specify what became the essence of these studies, i.e. the structure of the vertical boundary layer over Krakow, and as a result, they became an inspiration for further considerations. I often encounter mental abbreviations in the text, please pay attention to that.

The authors agree with the Reviewer's suggestions. In the final version of the article, these comments will be taken into account.

The specific geographical location of Krakow in the Vistula valley makes the meteorological conditions in the boundary layer of the atmosphere even more important here than in other regions. Therefore, empirical research on the structure of the boundary layer of the atmosphere has been carried out here for many years. The authors of the article wanted to highlight the contribution of the research at that time to the understanding of the structure of the boundary layer of the atmosphere, and in particular the use of sodar.

1.40 "wind speed and wind direction of the horizontal wind component (Ty and v)" – the order in brackets should be changed

Fixed

", the wind speed and wind direction of the vertical wiatru component (w)" – here, in turn, the symbol is missing; "wiatru" – – this is a Polish word; wind or the wind?

Fixed

153 DC – the abbreviation requires explanation in the text

In-text (DC – diffusion class)

Figure 5 – please apply the same scale to the three charts; "(c) context" – please specify it is "spectral background" (?), please describe the y-axis

The y-axis is described by dB. Spectral background is noise independent of reflection, noise, or other sources of noise.

I 186 "To Since....." – unclear ?

It's a matter of translation - corrected

176 " This process is shown in Figure 5a-c" – Please specify, does this example refer to a specific day or time?

Yes, a specific hour and a specific 5 minutes

This is an illustration of the spectrum elaboration method

187 The authors write, "... that each spectrum is characterized by a single number (parameter)" - what? And then they write "Attempts were made to determine as many parameters of the spectrum as possible". Please explain this.

See review no. 2

204 "The ASP C profile …" Please explain ASP C

See review no. 2

202-206 Two functions are presented, but one is used in this work. Please clarify the article and discuss only the results presented in the study. Many parameters characterizing ASP are also listed, but there is no clear explanation of their use in the study

See review no. 2

Figure 7 "Example of the regularity factor (RF) for characterization a (mean spectrum) between the SODAR data height value of 35-205 m and the PM10 concentration" - there is a match factor in the Figure and what does "a" mean - in the text this symbol was used for a parameter in the function; the measurement range 205 is also incorrect? What does " mean spectrum" mean - please clarify.

It should be a capital A previously defined. Mean factor is the average of the spectrum.

211 " This half adjustment factor was treated as a factor of the regularity of ASP" please clarify "the half adjustment factor".

Typographical error- this sentence has been removed

212-213 "A general feature that was noticed was that the transformed spectra were close to zero from a certain height. Therefore, two versions of the calculation were carried out: up to 19 heights (215 m) and up to 28 heights (305 m) "– these results are not shown. In Fig. 7 is up to 205? Please discuss only those results that are documented in the work, and others can be mentioned by referring to the literature.

Corrected Results are shown only for the data that is shown in the illustration

217-219 " The regularity factor for the characteristic was compared with the average concentration of PM10 from three automatic NEM stations near the SODAR location (on Krasinski Avenue, Bulwarowa and Bujaka Streets). For clarity, only one MF was shown, but ..."— please do not use mental abbreviations, but express precisely what characteristic? Please explain why the results from three measurement stations with different purposes: communication, industrial and urban background were averaged?

Three stations of different nature were chosen because they tried to approximate the average concentration of PM10 for Krakow and they had the most complete sequence of measurements. Corrected in text

219 "For clarity, only one MF was shown…" - before the RF symbol appeared, please specify

See review no. 2 and above

220" After analyzing many ASPs, the hypothesis suggesting that ASP were disturbed a few hours ago, after which the concentration of PM10 increased, was introduced" – please document this.

Maybe it was mispresented, but the whole work serves to show some clear relationships between the concentration of PM10 and the state of the atmosphere and the concentration of PM10 from 12 hours ago. Clarifications have been made in the text of the article.

229 "....it was decided that the meteorological RF wind characteristics should be added to the regularity coefficients A to I" - in I. 189, the term "parameters of spectrum" was used for A-I, so it is unclear whether these parameters are marked as regularity coefficients.

Yes, for the sake of uniformity of the determinations, it was decided to call meteorological parameters coefficients of regularity.

Please explain all symbols used in the equations (4) and (5).

Improved and standardized markings. RFch is the regression coefficient for the characteristic ch=A,...,M

Table 3 – what do the numbers mean in the statistical parameters?

There are their numeric identifiers

Figure 6 Example of spectrum and SODAR function  $\Phi$  for two cases: (a) irregularity to 18:00 UTC; (b) a good regularity to 23:00 UTC on 26 January 2018 – please verify whether this is a spectrum or an example of the ASP.

That's right, the signature is wrong. This is a graph of the regularity coefficient for parameter A compared to PM10 concentration – corrected.

220 – "After analyzing many ASPs, the hypothesis suggesting that ASP were disturbed a few hours ago, after which the concentration of PM10 increased, was introduced" - - It does not seem obvious from the data in Fig. 7, please provide a more detailed explanation

This hypothesis was based on the analysis of multiple graphs. Figure 7 is one of them. It happened so often that it became an inspiration for this work. Described in the text.

Eq. (3) should be corrected: twice b1cos(...)

Fixed

*Eq. (5) and the one following it do not match. Please number all equations consecutively*

Fixed.tags. See review no. 2 This is the same equation written in a different form

On behalf of all authors Leszek Ośródka Katowice, 27 December 2023